# Selenium and Zinc Biofortification of *Pleurotus eryngii* Mycelium and Fruiting Bodies as a Tool for Controlling Their Biological Activity

**DOI:** 10.3390/molecules25040889

**Published:** 2020-02-17

**Authors:** Piotr Zięba, Katarzyna Kała, Anna Włodarczyk, Agnieszka Szewczyk, Edward Kunicki, Agnieszka Sękara, Bożena Muszyńska

**Affiliations:** 1Department of Horticulture, Faculty of Biotechnology and Horticulture, University of Agriculture in Krakow, 31-425 Krakow, Poland; e.kunicki@urk.edu.pl; 2Department of Pharmaceutical Botany, Faculty of Pharmacy, Jagiellonian University Medical College, 30-688 Krakow, Poland; kat3kala@gmail.com (K.K.); annawlodarczyk1966@gmail.com (A.W.); agnieszka.szewczyk@uj.edu.pl (A.S.); muchon@poczta.fm (B.M.)

**Keywords:** biofortification, lovastatin, phenoloic compounds, *Pleurotus eryngii*, selenium, zinc

## Abstract

*Pleurotus eryngii* (DC:Fr.) Quel. is a cultivated mushroom of high culinary value and medicinal properties. Mycelium of *P. eryngii* is characterized by the ability of effective bio-elements absorption from growth media so it could be biofortified with trace elements with a functional activity in the human body. In this study, the ability of *P. eryngii* mycelia from in vitro cultures as well as fruiting bodies were investigated in terms of their effectiveness in zinc and selenium accumulation. The effect of Se and Zn biofortification on productivity, chemical compounds, and bio-elements content of *P. eryngii* was determined as well. To enhance Se and Zn content in *P. eryngii* fruiting bodies and mycelia, substrates were supplemented with sodium selenite, at a concentration of 50 mg L^−1^, zinc sulfate, and zinc hydro-aspartate at a concentration of 87.2 and 100.0 mg L^−1^, respectively. Mentioned Zn concentrations contained the same amount of zinc(II) ions, namely 20 mg L^−1^. The content of organic compounds include phenolic compounds and lovastatin, which were determined by a high-performance liquid chromatography with diode-array detector (HPLC-DAD) and reverse phase high-performance liquid chromatography (RP-HPLC) method with UV detection. The ability of *P. eryngii* to accumulate zinc and selenium from the culture medium was demonstrated. The degree of accumulation of zinc turned out to be different depending on the type of salt used. The present study also showed that conducting mycelium of *P. eryngii* in in vitro culture, with a higher content of zinc ions, can result in obtaining the materials with better antioxidant ability. The results of this study can be used to develop the composition of growing media, which ensures the production of biomass with the desired composition of elements.

## 1. Introduction

Polypore mushrooms from Basidomycota and *Pleurotus* genus are species with wide medicinal and cosmetological activity because of their immunostimulatory, anti-cancer, and anti-aging properties. King oyster mushroom, *Pleurotus eryngii* (DC.) Quél, is one of the best known species from this genus because of culinary value and pharmacological activity [1,2,3,4]. It is closely related to the oyster mushroom (*Pleurotus ostreatus* (Jacq. ex Fr.) P.Kumm), but it definitely differs concerning the fruiting body appearance. The sporocarp reaches a size of up to 15 cm. It is composed of cream-colored to a light brown hat with a lamellar type hymenophore going down and the thick, soft stalk, which is edible in contrast to other *Pleurotus* spp. [2]. *P. eryngii* is native to the regions ranging from the Atlantic coast of Europe through the Mediterranean area to Kazakhstan and India. In Poland, it is a rare species with a protected status. The distinguishing feature of *P. eryngii* is the parasitic behaviour. In nature, it occurs as an optional parasite of *Eryngium campestre* L., as well as on other Apiaceae, such as *Elaeoselinum asclepium* (L.) Bertol and *Ferula communis* L. [5,6].

*Pleurotus eryngii* is cultivated in organic substrates similar to those used for oyster mushroom, supplemented high-protein waste, e.g., rice bran. It is characterized by great potential of absorption nutrients from lignocellulose residues through an effective ligninolytic enzyme system (laccase, Mn-oxidizing peroxidases, and aryl-alcohol oxidase). Due to this ability, *P. eryngii* is used in many biotechnological processes including food production, biotransformation of raw plant materials to feed, bio-pulping, and bio-bleaching of paper pulp, or bioremediation of soil and industrial waters [7]. It reacts with extreme sensitivity to changes in growth conditions so some research were performed to modify the substrate composition to increase the level of chemical compounds. Generally, substrate additives, especially supplements rich in nitrogen and carbohydrates, are widely recognized and accepted in agricultural practice because of a substantial impact on mushroom yield and quality [8,9]. The mycelium produced in the liquid culture media have become a viable alternative for receiving a raw material for biomedical use, food, and enzyme production. The liquid culture technology was also adapted to the production of mycelia used for substrate spawning in cultivating fruit bodies [9]. The use of in vitro cultures leads to an increase in biomass, a reduction of production duration, and a stable production of raw material with a predictable amount functional metabolites [10]. The liquid medium composition, pH, temperature, time of cultivation, etc., have an important influence on biomass of the mycelium and its physicochemical and nutritional properties, but generally mycelia maintain the functional components of the parent mushroom, such as phenolic acids and ergosterol. The anti-inflammatory properties could be used in nutraceutical or pharmaceutical formulations [11,12]. Such formulations have been developed based on the in vitro mycelium cultures of the most important species like *P. ostreatus*, *P. diamor*, and *Agaricus bisporus* [10,11,12,13]. A relatively new approach is supplementation of growing medium with bio-elements employed for the cultivated mushrooms not only to improve their agronomic potential in terms of yield and quality but also to functional food and diet supplements production [14,15,16,17].

The *P. eryngii* fruiting body of this species has a nice flavor and rich taste, similar to *Boletus edulis* Bull. Sporocarp contains almost 20% of protein in dry matter and up to 4% fat, which makes it more nutritious when compared to most of the cultivated mushrooms [3,4]. Generally, mushroom species contain from 3 to 15 phenolic compounds, and gallic, and protocatechuic acids are the most commonly reported [18]. All pohenolics are used as indicators of the antioxidant capacity, which is linked to their hydroxyl groups [19]. Gallic acid is known to activate diverse pharmacological and biochemical effects in humans including strong anti-cancer, antioxidant, and anti-inflammatory factors [20]. In addition, it elevated the levels of enzymatic and nonenzymatic antioxidants [21]. *Pleurotus eryngii* is a significant source of lovastatin, also known as monocline k, which is a naturally occurring compound, and the hypolipidemic agent acts through the inhibition of the HMG-CoA reductase [22]. Lovastatin not only reduces blood cholesterol, but it also has anti-inflammatory, anticoagulation, antioxidative, anti-fungal, and anti-carcinogenic effects [23,24]. Acidic and alkalic-extractable zinc polysaccharides extracted from *P. eryngii* mycelia have demonstrated hepatoprotective and antioxidant effects [25]. *P. eryngii* contains the highest total glucan concentrations among *Pleurotus* spp., concentrated mainly in stalk and the highest α-glucans proportion. Another compound, immunostimulatory β-glucan—pleuran has antioxidant properties, which, combined with high taste values, makes the fruiting bodies of this species one of the most valuable cultivated mushrooms [1]. Generally, mushrooms contain higher amounts of bio-elements than agricultural crops, vegetables, and fruits because of the presence of effective mechanisms of their active uptake [26]. Bio-elements accumulation in *Pleurotus* mycelium should be considered as a potential alternative to produce non-animal food sources of essential elements.

Selenium is an essential component of several major metabolic pathways in human cells, such as thyroid hormone metabolism, antioxidant defense, immune function, reproduction, and viral inhibition. Selenium is involved in biosynthesis of selenoenzymes and selenoproteins, as glutathione peroxidases, iodothyronine 5′-deiodases, thioredoxin reductases, selenoprotein P, and selenoprotein W. Selenoprotein P is a transport protein maintaining Se homeostasis while selenoprotein W is involved in the antioxidant defense of cardiac and skeletal muscle as well as in cell cycle progression [27,28]. Zinc is a strong antioxidant because it acts as a cofactor of superoxide dismutase and many other enzymes [29]. Zn stabilizes the molecular structure of subcellular organelles and their membranes and it is required for metabolism of nucleic acids, proteins, carbohydrates, lipids, and secondary metabolites, which affects cell division, growth, and repair [30]. Zn and Se deficiency resulted in decreased antioxidant capacity, whereas supplementation with these elements significantly improved the antioxidant status in animal testes [31]. The mushrooms, including *Pleurotus* spp., have the capacity to absorb, accumulate, and transform Se inorganic compounds into organic compounds [32], but the chemical forms of metal salts used for supplementation can affect the mycelial growth and fruit bodies production, and its chemical composition as well.

The aim of the presented work was to determine the content of bio-elements and organic compounds in *P. eryngii* mycelium from in vitro cultures and fruiting bodies grown in media/substrates fortified with zinc and selenium salts. We hypothesized that (i) Zn and Se affect the yield and chemical composition of mycelium/fruiting body in different ways, (ii) mycelium from in vitro culture could be used as a comparative or alternative model to obtain raw material with medicinal properties, and (iii) diversified biofortification allow us to obtain raw material with chemical compounds targeted for utilization by food, pharmaceutical, veterinary, or cosmetology industries.

## 2. Results

### 2.1. Organic Compounds Total Phenolics and 2,2-Diphenyl-1-picrylhydrazyl ( DPPH) Scavenging Activity Analysis

The yield of *P. eryngii* fruiting bodies was of 12.28 g dry weight (d.w.) per 100 g^−1^ substrate. Fortification with Zn sulfate and Zn hydroaspartate did not cause a decrease of the yield, but Se application affected in two-fold yield reduction was compared to the remaining trials. The mycelium yield was, on average, 6.09 g d.w. per 1 dm^3^ of liquid Oddoux medium. Application of Zn sulfate and Zn hydroaspartate significantly increased the mycelium yield as compared to the control while selenite did not affect *P. eryngii* mycelium productivity (Figure 1).

*Pleurotus eryngii* mycelium and fruiting bodies were differentiated with respect to organic compounds levels and showed a different reaction against Zn and Se biofortification (Table 1). Supplementation of growth substrate with Zn sulfate caused a significant increase of *p*-hydroxybenzoic acid and cinnamic acid in *P. eryngii* fruiting bodies when compared to the control. Supplementation with selenite resulted in a decrease of phenylalanine, 3, 4-dihydroxyphenylacetic acid, and *p*-hydroxybenzoic acid, whereas Zn hydroaspartate implementation caused a decrease in all investigated organic acid contents when compared to the control. Supplementation of in vitro growth medium with Se and Zn significantly increased phenylalanine and gallic acid contents, whereas Zn sulfate and Zn hydroaspartate positively affected 3, 4-dihydroxyphenylacetic acid content in mycelium as compared to the control. Protocatechuic, syringic, and cinnamic acids were not detected in mycelium while gallic acid was not detected in fruiting bodies of *P. eryngii*. Generally, phenylalanine content in mycelium was 102% higher than those determined in fruiting bodies, but 3, 4-dihydroxyphenylacetic and *p*-hydroxybenzoic acid contents were lower at 88% and 91%, respectively.

Lovastatin was determined only in *P. eryngii* mycelium. Supplementation with Se and Zn strongly decreased lovastatin content in *P. eryngii* mycelium. We determined 27.02 mg 100 g*^−^*^1^ d.w. of lovastatin in control treatment, while 1.02 mg 100 g*^−^*^1^ d.w. in average in Se and Zn supplemented treatments. Total phenolics content was a little higher in mycelium 3.07 mg 100 g d.w., average for treatments) than in fruiting bodies 2.21 mg 100 g*^−^*^1^ d.w. Significant differences between treatments were determined only for fruiting bodies. Zn sulfate fortification caused a significant increase of these compounds. 2,2-Diphenyl-1-picrylhydrazyl (DPPH˙) scavenging activity was 36.7% and 48.0%, on average, for fruiting bodies and mycelium, respectively. Supplementation with selenite and Zn sulfate of growth substrate significantly increased DPPH˙ scavenging activity in fruiting bodies. Supplementation of in vitro medium with Zn sulfate and Zn hydroaspartate decreased antioxidant activity in mycelium.

Pearson’s correlation coefficients between Zn and Se and organic compounds in fruiting bodies showed a positive effect of Zn on phenylalanine, *p*-hydroxybenzoic acid, and cinnamic acid content and a negative effect on total phenolics and DPPH˙ scavenging activity (Table 2). Selenium was positively correlated with syringic acid. Correlations determined for mycelium were positive in the case of Zn for phenylalanine, gallic, protocatechuic, and *p*-hydroxybenzoic acids, while being negative for lovastatin and DPPH˙ scavenging activity. In the case of Se, positive correlation was determined only for DPPH˙ scavenging activity, while a negative correlation was determined for gallic, 3,4-dihydroxyphenylacetic, and *p*-hydroxybenzoic acids as well as for lovastatin.

The principal component analysis (PCA) was performed to compare mycelium and fruiting bodies in organic compounds’ composition. The first two principal components (PCs) explained 90.95% of the total variation in the data for fruiting bodies and 94.87% for mycelium (Figure 2), which reflects the complexity of the relationships among the treatments and the organic acids’ content. All traits’ vector lengths are of almost maximum length. Therefore, their effects are significant. Acute angles (<90 °) between vectors representing syringic, protocatechuic, 3,4-dihydroxyphenylacetic acids, and phenylalanine as well as cinnamic and *p*-hydroxybenzoic acids depict the positive correlation between these characteristics for fruiting bodies. A positive correlation was also shown between all organic compounds determined in the mycelium.

### 2.2. Bioelements Analysis

Bioelement content was differentiated depending on source of *P. eryngii* material as well as Se and Zn supplementation of substrate and growth medium (Table 3). Potassium content was three-fold higher in fruiting bodies than in mycelium. Biofortification with Zn sulfate and Zn hydro-aspartate decreased K accumulation in fruiting bodies, while supplementation with Se, followed by Zn, decreased significantly K content in mycelium. The mycelium supplemented with Se contained a two-fold lower level of calcium than the control. *Pleurotus eryngii* fruiting bodies grown in substrate fortified with Se and Zn hydroaspartate accumulated more magnesium than in the control. Mycelium supplemented with Zn sulfate and Zn hydro-aspartate also contained a significantly higher Mg level, but those biofortified with Se contained a lower Mg level as compared to the control. Supplementation with Se and Zn increased Ca content in fruiting bodies, while enrichment of liquid in vitro medium with Se decreased Ca content in mycelium as compared to the control. Zinc-content in fruiting bodies increased significantly only as an effect of Zn hydro-aspartate supplementation of growth medium while Zn content in mycelium was 19-fold and 12-fold higher than in the control after addition of Zn sulfate and Zn hydroaspartate, respectively, to liquid medium.

Application of Se and Zn to growth substrates significantly increased iron content in fruiting bodies of *P. eryngii*, while application of Se to liquid solution increased, and Zn salts decreased Fe content in mycelium as compared to the control. Mycelium contained 27.1 mg 100 g^−1^ d.w., while fruiting bodies had 4.8 mg 100 g^−1^ d.w., on average for treatments. The chlorine level was not diversified in fruiting bodies from experimental treatment, while Se application to liquid medium caused an increase of its level in mycelium as compared to the control. Mycelium contained about 11-folds higher level of Cl when compared to fruiting bodies. Rubidium content in *P. eryngii* fruiting bodies and mycelium decreased as a result of biofortification of growth substrate and liquid medium. The mean for treatments’ content of Rb in fruiting bodies was 1.41, while, in mycelium, it was −0.22 mg 100 g^−1^ d.w. Biofortification decreased copper content in fruiting bodies, but increased in mycelium but only in the case of Zn salts. Generally, fruiting bodies contained three-fold more Cu than mycelium. Se biofortification was effective in the case of fruiting bodies and mycelium. Both tissues were significantly enriched in this element even though mycelium accumulated a 15-fold higher amount of this element (mean for treatments). Manganese content in *P. eryngii* fruiting bodies was not affected by substrate supplementation, while it decreased in mycelium grown in liquid medium enriched with Se and Zn. Fruiting bodies contained a mean of 0.88 mg 100 g^−1^ d.w, and mycelium from 4.46 to 9.98 mg 100 g^−1^ d.w. of Mn. Supplementation of substrate with Zn sulfate decreased nickel content in fruiting bodies. There were no noted significant differences between the remaining treatments for Ni content in fruiting bodies and mycelium. No statistical differences were also noted for *P. eryngii* fruiting bodies and mycelium treatments regarding chrome, bromine, and strontium content.

The principal component analysis (PCA) was performed to compare mycelium and fruiting bodies in bioelements’ composition. The first two principal components (PCs) explained only 55.05% of the total variation in the data for fruiting bodies, but 72.7% for mycelium. Significant relationships among the bioelements analyzed were found, although they varied in *P. eryngii* fruiting bodies and mycelium (Figure 3). The analysis of fruiting bodies showed inverse correlations between Se and Cu, and positive correlations for Se and Ca, Mg, Cr, and Sr. The Zn was positively correlated with K and Rb, but negatively correlated with Ca and Cr. In the case of mycelium, Se was positively correlated with the contents of Fe and Br, whereas negative associations occurred for K, Ca, Mg, Mn, Zn, Rb, and Sr. The Zn enrichment had a significantly negative effect on Fe, Ni, Se, and Br content, but had a positive effect on Cu and Mg levels.

## 3. Discussion

The results of this study significantly contribute to the rise of information on the species consumed as fresh mushrooms and the possibility of bio-elements enriched mycelium and fruiting bodies production as a source of bioactive compounds. Munoz et al. [33] determined that Se specifically binds to chitin in the cell walls of *P. ostreatus*. Moreover, lower Se concentration (2.5 mg L^−1^) stimulated the growth of the mushroom, while higher concentration (5 mg L^−1^) had an inhibitory effect. Kim et al. [34] investigated the growth and enzyme activity in *P. eryngii* grown in medium supplemented with 1, 10, 100, 1000, and 10,000 µM of sodium selenite. Mycelial growth was increased at lower Se levels, but declined significantly at 1000 and 10,000 µM of Se as a result of Se toxicity. In the present study, Se did not affect the in vitro mycelium yield, but significantly decreased fruiting bodies’ weight despite the concentration of this element being higher in supplemented mycelium. Se-mediated stimulating or toxic response in *P. eryngii* growth could be differentiated during vegetative (mycelium) and generative (fruiting bodies) stage of development as well as growth conditions (liquid medium vs. growth substrate). As a mechanism of a metal mediated growth response, Stajic et al. [7] and Kim et al. [34] demonstrated increased laccase activity at low levels of Se, Fe, and Zn and reduced activity at high levels of these elements in *P. eryngii*, *P. ostreatus*, and *P. pulmonarius*. Fungal laccases control various functions as degradation of lignin and many xenobiotic compounds, morphogenesis, stress, and pathogen defense indirectly affect fungal growth and development. They could be involved in Se and Zn response in *P. eryngii* yielding in the present research. In terms of mushroom productivity, Zn supplementation seems to be the most promising approach in liquid media cultures. Enrichment of liquid medium with Se leads to stable mycelium production with respect to dry weight while Se application to a growing substrate causes a decrease of fruit bodies’ production.

### 3.1. Organic Compound—Total Phenolics, Lovastatin, and DPPH˙ Scavenging Activity Analysis

In the present study, gallic acid content in *P. eryngii* mycelium was negatively correlated with 2,2-Diphenyl-1-picrylhydrazyl (DPPH) scavenging activity (r = −0.962; *p* = 0.000) even though it does not exclude its positive action in the human body. Moreover, zinc and, in a lesser degree, selenium supplementation positively affected gallic acid content in *P. eryngii* mycelium, while this acid was not detected in fruiting bodies. According to Carrasco-Gonzalez et al. [21], coumaric and ferulic acid contents in *P. ostreatus* fruiting bodies were positively affected through Se biofortification.

We determined seven phenolic compounds with phenylalanine as a dominating agent in fruiting bodies and mycelium of *P. eryngii*. In turn, 12 phenolic acids, namely gallic, protocatechuic, chlorogenic, caffeic, vanillin, ferulic, naringin, resveratrol, naringenin, hesperetin, formononetin, and biochanin-A were detected in fruiting bodies of *P. citrinopileatus* [18]. Gallic acid, protocatechuic acid, chlorogenic acid, vanillin, ferulic acid, naringin, naringenin, hesperetin, formononetin, and biochanin-A were detected from acetonitrile and 0.1 N hydrochloric acid solvent extract of *P. eryngii* fruiting bodies, and protocatechuic acid was determined in the highest concentration [35]. We performed principal component analysis (PCA) analyses to compare mycelium and fruiting bodies in organic compounds’ composition. Because the first two PCs explained more than 90% of the total variation in the data for fruiting bodies and mycelium, we can conclude that the Se and Zn supplementation and the organic acids’ content have complex relationships. According to Yan and Chang [31], the boxplot model fits well if the first two PC’s should reflect more than 60% of the total variation. Basing on the cited and presented results, it can be stated that the phenolic acids show great differentiation among *Pleurotus* species even within the same species because of genetic differences, substrate composition, growing conditions, or mushroom raw material. Se and Zn supplementation can be used as a tool to control phenolic acids’ level in *P. eryngii* mycelium and fruiting bodies. In the present study, this relationship was clearly demonstrated for gallic acid accumulation in *P. eryngii* mycelium, which is positively affected by Zn and Se addition to liquid solutions.

Phenolic compounds have been reported to be the major antioxidants determined in mushrooms. Moreover, radical scavenging activity has a strong correlation with the phenolic content in *Pleurotus* spp. [36]. Our results confirmed this relationship for fruiting bodies with a correlation coefficient value of r = 0.751, *p* = 0.001, while, for mycelium, it was non-significant (r = 0.234, *p* = 0.383, even though total phenolics content was higher in mycelium as compared to fruiting bodies. It can be explained that the slight effect of Zn and Se supplementation on total phenolics in mycelium and a significant decrease of mycelium DPPH radical scavenging activity by Zn salts’ supplementation. In the present study, total phenolics in *P. eryngii* samples determined by the Folin-Ciocalteu assay was higher than reported by Li and Shah [35], namely 1090.42 ± 13.77 µg GAE g^−1^ d.w., but lower than the results by Reis et al. [37] for methanolic extracts from fruiting bodies (7.14 ± 2.01 mg GAE g^−1^) as well as mycelium (9.11 ± 0.23 mg GAE g^−1^). This may be due to differences in growth medium, growing conditions, extraction solvents, and the conditions used.

Atli and Yamac [38] screened 136 macro-fungi isolates from the Basidiomycetes for lovastatin production and only six of them were found to be lovastatin producers, *Omphalotus olearius*, and *P. ostreatus*, which is the most effective. Lovastatin seems to be an uncommon compound in higher mushrooms, so the demonstration of *P. eryngii* non-supplemented mycelium effectiveness in lovastatin synthesis is an important achievement of the present research. In addition, in our research, we proved that modification of Oddoux medium by adding extra Zn and Se salts, intensely decreased lovastin production of mycelium, but mechanisms of this phenomenon need future investigation. Anyway, the decision on Zn and Se supplementation should be taken with the awareness that it could negatively affect lovastatin content in mycelium from in vitro cultures. There is also a need of future investigations on the liquid media formulations leading to increased lovastatin content in *P. eryngii* mycelium.

Generally, Se is considered to be an essential nutritional trace element of strong antioxidant function [39]. According to Kim et al. [40], Se at concentrations above 100 µM leads to decreases of antioxidative functions of peroxidase and laccase in the *Pleurotus eryngii*. *Pleurotus ostreatus* enrichment with Se also affected the antioxidant capacity, and, particularly, the methanolic extract obtained from the second flush of fruiting bodies had the best antioxidant activity [21]. In this study, the DPPH˙ scavenging activity of fruiting bodies was significantly increased by Se addition to the growth substrate, which indicates the potential use of Se-enriched *P. eryngii* as antioxidants in diet. Fasoranti et al. [17] determined that antioxidant activity of the ethanol extracts of Se-enriched mushrooms was significantly higher than the non-enriched *Pleurotus* species. Selenium has strong antioxidant activity. Moreover, the mechanism of DPPH˙ scavenging power may be an incorporation of selenyl group (SeH) or seleno-acid ester to three-dimensional structure of polysaccharide that changed the hydrogen atom-donating capacity [41]. In general, it can be suspected that the significantly improved antioxidant profile of mushrooms was enriched with Se, which was also stated by Kaur et al. [42].

### 3.2. Bioelements Analysis

Mycelia of *P. eryngii* show the ability of trace elements’ accumulation in amounts depending on the metal source in medium [43]. Some of them are essential for fungal metabolism as enzyme components or enzyme activity modulators. According to Gogavekar et al. [44], the mean metal concentration in *P. ostreatus* fruiting bodies was in the order: Ca  >  Fe  >  Mg  >  Na  > K  >  Zn  >  P  >  Ni  >  Mn  >  Pb  >  Cu  >  Cr  >  Co. *Pleurotus* spp. could be an important source of nutritionally valuable minerals because they contained high amounts of K, Mg, and Ca [45]. Moreover, enrichment of growth medium with Se and Zn was effective in biofortification in this element including both mycelium and fruiting bodies. In the present research, the increase of Se concentration in the liquid medium led to an increase of its content in mycelium of *P. eryngii*. Poursaeid et al. [46] determined that the ability of Zn bioaccumulation in the mycelia of *P. florida* was much higher than in the fruiting bodies. The enrichment of growth substrate in minerals, namely Zn, Li, and Fe, promoted a decrease in the content of Fe in *P. ostreatus* fruiting bodies. Moreover, *P. ostreatus* enriched with Fe, Zn, or Li provided minerals such as K, P, Fe, Zn, Li, and Cu. In addition, no heavy metals such as Ni, Cr, and Cd were detected, and only low levels of Pb and Al were observed [47]. Bioelements are involved in all metabolic reactions, transmission of nerve impulses, regulation of water and salt balance, and many other processes crucial for proper functioning of the human body [48]. *P. eryngii* is a good source of minerals including potassium, phosphorus, magnesium, manganese, zinc, and calcium. Moreover, the content of crucial elements could be modified through growth medium enrichment. We can recommend that addition of Zn salts to growth substrate lead to increased K and Rb content, while supplementation with Se increases Br, Sr, Mg, Cr, Ca, and Fe. We can also formulate the recommendations for controlling minerals in mycelium through Zn and Se supplementation. Essentially, the addition of Zn to liquid medium can increase accumulation of Cu in mycelium, while the addition of Se salt can increase Br, Fe, and Cl. These results show complex bio-elements’ interactions in mushroom fruit bodies and mycelium, which is reported by Siwulski et al. [49]. The amount of nutrients recommended per day for Americans four years of age or older by FDA [50] are for Ca 1000 mg, Cr 120 µg, Cu 2 mg, Fe 18 mg, Mg 400 mg, Mn 2 mg, P 1000 mg, K 3500 mg, Se 70 µg, Na 2400 mg, and Zn 5 mg. The present study confirmed that *P. eryngii* has beneficial effects from a nutritional point of view K:Na ratio such as low Na and high K content. K affects the reduction of blood pressure, lowers the risk of cardiovascular disease, stroke, and coronary heart disease as well [51]. We demonstrated that 1 g of dry *P. eryngii* mycelium supplemented with Se can provide 268% of recommended daily intake, while it was supplemented with Zn sulfate 58% of recommended daily intake. Therefore, supplemented mycelium could be used as a diet supplement providing bioelements as Zn and Se with additional biological active compounds that can be find in mycelium. In our growing experiment, the effect of Zn supplementation was vestigial and only a slightly increased amount of it in fruiting bodies of *P. eryngii.* Supplementation fruiting bodies with Se was more effective, but still 20 times less effective than mycelium and a negatively affected yield of mushrooms. Despite this supplementation, other elements’ absorption substantially change organic compounds in fruiting bodies and mycelium. More specific research is needed to understand these phenomena. According to Estrada et al. [52], selenium-enriched mushrooms would supply more than 20% of the daily intake. They could be considered an excellent source of selenium. Our results demonstrated that both fruiting bodies and mycelium of *P. eryngii* can be considered as functional food or raw material enriched in essential nutrients, which can be widely used in food supplementation.

Despite total bioelements content, the amount provided by mushroom consumption is not always enough to meet nutritional requirements of humans if the bioavailability is low, as for non-heme Fe or Zn. According to Kalac and Svoboda [48], the bioavailability of iron in mushrooms is considerably high and human body can absorb up to 90% of the available form.

## 4. Materials and Methods

### 4.1. Reagents and Standards

Zinc hydroaspartate (C_8_H_12_N_2_O_8_Zn) was obtained from Farmapol (Poznań, Poland), zinc sulfate (ZnSO_4_⋅7H_2_O) from OUM-7, (Łódź, Poland) and sodium selenite (Na_2_SeO_3_) from Sigma-Aldrich (St. Louis, MO, USA). Water (four times distilled) with a conductivity of less than 1 µS cm^−1^ was obtained using an S2-97A2 distillation apparatus (ChemLand, Stargard Szczecin, Poland).

Standards of organic compounds: *p*-hydroxybenzoic acids were obtained from Fluka (Chemie AG, Germany), and gallic acid, 3-4-dihydroxyphenylacetic acid, syringc acid, cinnamic acid, protocatechuic acid, lovastatin, 2,2-Diphenyl-1-picrylhydrazyl (DPPH˙) radical, Folin & Ciocalteu’s phenol reagent from Sigma-Aldrich (St. Louis, MO, USA). HPLC-grade methanol, phosphoric acid, acetic acid, and acetonitrile from Merck (Darmstadt, Germany). Concentrated HNO_3_ Suprapur^®^, and H_2_O_2_ Suprapur^®^ from Merck (Darmstad, Germany).

Chemicals for Oddoux medium: glucose, maltose extract, casein hydrolysate, L-asparagine, adenine, and yeast extract were purchased from Sigma-Aldrich (St. Louis, MO, USA). NH_4_Cl, KH_2_PO_4_, MgSO_4_⋅7H_2_O, CaCl_2_⋅6H_2_O, FeCl_3_, MnSO_4_⋅H_2_O, from PPH Golpharm (Kraków, Poland). Growing medium, namely wheat straw and beech sawdust, were bought from a local producer as a homogeneous batch.

HClO solution was manufactured by Unilever (Nyírbátor, Hungary), n-hexane, chloroform, were purchased from Merck (Darmstad, Germany).

### 4.2. Pleurotus Eryngii Materials

The *P. eryngii* explants for the generation of the in vitro cultures were provided courtesy of the prof. Marek Siwulski, Poznań University of Life Sciences. Representative voucher specimens were deposited at the Department of Pharmaceutical Botany, Jagiellonian University Medical College, Kraków, Poland. In order to conduct the experiment*,* in vitro cultures of *P. eryngii* were grown on a modified liquid medium with composition according to Oddoux.

### 4.3. In Vitro Cultures of Pleurotus eryngii

To achieve a maximum efficiency in mushroom biomass growth, the mushroom cultures were transferred to the modified liquid Oddoux medium (starting inoculum transplanted from the solid medium culture was 0.1 g). To prepare the experimental cultures, the obtained biomass was passed into Erlenmeyer’s flasks (500 mL) containing the liquid medium (250 mL) and conducted at 25 ± 2 °C under 16 h of lighting (11.5 µmol s^−1^ m^−2^) and 8 h of darkness. The biomass was obtained from the cultures grown on the Oddoux medium (control), and on the same medium, but with the addition of sodium selenite at a concentration of 50 mg L^−1^ (0.00029 mol L^−1^), zinc hydroaspartate at a concentration of 100 mg L^−1^ (0.00027 mol/L), and zinc sulfate (0.000304 mol L^−1^) at a concentration of 87.23 mg L^−1^. The applied concentrations of both compounds contained the same content of zinc (II) ions (20 mg L^−1^). After four weeks, since the initiation of in vitro cultures on liquid medium, the biomass was separated from the medium and rinsed three times with four-fold distilled water. The resulting biomass was frozen and then dried via lyophilization (lyophilizer Freezone 4.5, Labconco, temperature: −40 °C).

### 4.4. Fruiting Bodies of Pleurotus eryngii

Mushrooms were grown in 3-L glass jars, on beech sawdust, and grinded wheat straw (1:1 *v*/*v*) moisturized with distilled water to the content of 65% ± 1%. Zinc and selenium compounds’ concentration was the same per liter as in in vitro cultures. Jars were filled with the mix of salts and growing medium, and sterilized at 121 °C for 1.5 h. The cooling substrate was inoculated with previously prepared wheat grain spawn of *P. eryngii* (3% of substrate weight) in a cavity made in the center of a jar. Jars were incubated at 24 ± 2 °C in dark for four weeks until mycelium fully colonized the substrate. After then, jars were put in a growing room with a temperature of 18 ± 1 °C, humidity of 95 ± 3%, and light intensity of 11.5 µmol s^−1^ m^−2^ with a 12 h photoperiod. Each treatment consisted of four jars. Fruit bodies were harvested in a stage of market maturity and dried in a laboratory dryer at 40 °C.

### 4.5. Mushroom Extracts

For the preparation of *P. eryngii* methanol extracts, the lyophilized mushroom materials such as the fruiting bodies and biomass from the in vitro culture were portioned and weighed (5 g of each sample), then ground in a mortar, and subjected to extraction with petroleum ether in percolators in order to remove the lipid fraction, according to the procedure developed by Kała et al. [53]. The remaining degreased biomass was dried and again subjected to extraction with methanol in a percolator for 24 h (kept in the dark). The obtained extracts were concentrated by distillation in a vacuum evaporator (Büchi, Germany) under reduced pressure (200 mbar) at 40 °C. The resultant extracts were dissolved in methanol (1 g of dry extract to 1 mL of methanol) and then filtered through bacteriological 0.2 µm syringe filters. Then, they were diluted to desired concentrations. The prepared extracts were stored at 4 °C until use.

### 4.6. Organic Compounds Analysis—Lovastatin

Chromatographic separation was performed using a high-performance liquid chromatography (HPLC) analyzer (Merck Hitachi). The process was carried out in an isocratic system with a mobile phase of constant composition. The apparatus was equipped with a ultraviolet (UV) detector (λ = 238 nm), a column (Purospher RP18 14 × 200 mm, 5 µm), and a lamp (L7100). During each measurement, 20 µL of the analyzed sample was injected, and the measurement was performed within 15 min. All measurements were carried out using a previously prepared developing system (acetonitrile and 0.1% phosphoric acid in the ratio of 60:40 (*v*/*v*)). The retention time of the standard substance was 12.75 min.

### 4.7. Organic Compounds Analysis—Phenolic Compounds

Methanolic extract was evaporated to dryness (Buchi evaporator, Germany) under a pressure of 200 mBa at 40 °C. The residues were quantitatively dissolved in methanol (1.5 mL) and filtered through a Millipore Millex–GP, 0.22 µm.

The resultant extracts were analyzed for their contents of phenolic acids by the high-performance liquid chromatography with diode-array detector (HPLC-DAD) method. These analyses were carried out according to the procedure developed by Kała et al. [53], where methanolic extracts from 12 species of fruiting bodies of edible mushrooms were analyzed using the HPLC method with some modifications. HPLC analyses were conducted using an HPLC VWR Hitachi-Merck apparatus: L-2200 autosampler, L-2130 pump, RP-18e LiChrospher (4 mm × 250 mm, 5 µm) column thermo-stated at 25 °C, L-2350 column oven, and L-2455 diode array detector at the UV range of 200–400 nm. The mobile phase consisted of solvent A: methanol/0.5% acetic acid 1:4 (*v*/*v*) and solvent B: methanol. The gradient was as follows: 100:0 for 0–25 min, 70:30 for 35 min, 50:50 for 45 min, 0:100 for 50–55 min, and 100:0 for 57–67 min. The comparison of UV spectra and retention times with standard compounds enabled the identification of phenolic acids present in analyzed samples. The quantitative analysis of free phenolic acids was performed with the use of a calibration curve with the assumption of the linear size of the area under the peak and the concentration of the reference standard.

### 4.8. Scavenging Activity Analysis (%DPPH˙)

0.1 mL of mushrooms methanolic extracts was mixed with 0.1 mM 2,2-Diphenyl-1-picrylhydrazyl (DPPH) dissolved with 4.9 mL of 100% methanol. The mixture was shaken and kept in dark for 45 min. The absorbance was measured at 517 nm using a Ultraviolet-Visible Spectroscopy UV-VIS Helios Beta spectrophotometer (Termo Fisher Scientifc Inc., Waltham, MA, USA). DPPH radical scavenging activity was calculated using the formula: AA (%) = [(A0−A1)/A0] × 100, where AA is the antioxidant activity, A0 is the absorbance of the reference solution, and A1 is the absorbance of the test solution.

### 4.9. Total Phenol Content

Total phenolic content was estimated using the modified Folin-Ciocalteu colorimetric method. Additionally, 0.1 mL of mushrooms’ methanolic extracts was mixed with 2 mL of sodium carbonate. After the next 2 min, 0.1 mL of Folin-Ciocalteu’s reagent, mixed with deionised water (1:1 *v/v*), was added. The absorbance of the resulting blue colour was measured at 750 nm using the UV-VIS Helios Beta spectrophotometer against a reference solution. The results are expressed as gallic acid equivalents (GAE).

### 4.10. Bioelements Analysis

The amount of elements in the fruiting bodies and mycelium of *P. eryngii* was determined using the total reflection x-ray fluorescence spectrometry (TXRF) method. The TXRF method was chosen for this study due to several significant aspects such as high sensitivity, precision, accuracy, and repeatability of analyses as well as the possibility to achieve a faster measurement with a very large number of samples. The optimization of mineralization conditions for analysis of samples in combination with the TXRF method enabled optimal analysis of bioelements in the fruiting bodies, and mycelia cultures of *P. eryngii.*

For this analysis, 0.2 g of samples of lyophilized mushroom material were weighed, with an accuracy of 0.01 g, and were transferred to Teflon vessels to which 2 mL of H_2_O_2_ solution (30%) and 4 mL of concentrated HNO_3_ solution (65%) were added. Mineralization was carried out in Magnum II microwave apparatus (ERTEC) in three stages of 10 min each at a power of 70% and 100%, respectively, while maintaining the temperature of the device at 290 °C. After mineralization, the solutions were transferred to quartz evaporators and evaporated on a heating plate at 150 °C to remove excess reagents and water. The residue obtained after evaporation was quantitatively transferred to 10 mL of volumetric flasks with four-times-distilled water. To analyze the composition of bioelements such as K, Ca, Mn, Fe, Cu, Zn, and Se in the prepared test samples, 1000 ppm gallium was used as an internal standard. The composition of elements was measured using a TXRF spectrometer Nanohunter II (Rigaku) equipped with an X-ray tube containing a molybdenum anode at 50 kV for 1000 s.

The highest detection limits (LOD) are obtained for heavy elements: potassium and calcium above 1 mg/kg for light elements. The remaining LOD values oscillate below 1 mg/kg (0, 0.09, 012, 0.22, 0.29, 0.38, 0.38, 0.45, 046, and 0.99 for Rb, Se, Sr, Ni, Cu, Zn, Fe, Mn, Cr, and Mg respectively).

### 4.11. Statistical Analysis

Data were subjected to analysis of variance (ANOVA) followed by Tukey’s HSD tests using the Statistica 12.0 software package (StatSoft Inc., Tulsa, OK, USA). Correlation coefficient r was calculated to determine the relation between Zn, Se, and each of the chemical traits. Experimental data were also processed for a principal component analysis (PCA) in order to evaluate the existing relationships with original variables.

## 5. Conclusions

*P. eryngii* is found to be a rich source of important major minerals like K, Mg, Ca, Fe, and some minor minerals. This species contains a significant amount of secondary metabolites like phenolic acids, polyphenols, and lovastatin being considered as compounds having a wide influence on human body functions. Therefore, *P. eryngii* fruiting bodies and mycelium can be used as a raw material for diet supplements production and for pharmaceutical industry. *P. eryngii* fruiting bodies consumption provides a wide range of major and minor nutrients, so it can be treated as a functional food to combat various deficiency diseases and malnutrition. Enrichment of growing media in Zn and Se salts leads to higher content of these metals in the mycelium and fruiting bodies with a diverse effect regarding other chemical constituents. Due to the fact that fruiting bodies’ production takes a long time to complete, and the production is difficult to control due to the use of different agricultural residues, the mycelium may be an economic and safe alternative for the production of raw material for different industries. Careful selection of dose, form, and the way of Zn and Se application enable us to produce *P. eryngii* fruiting bodies or mycelium with predictable amounts of these elements. Considering the complex effect of supplementation on mushrooms’ chemical composition, precise formulations for growing media composition should be formulated to produce raw material standarised with respect to particular elements or chemical compounds.

## Figures and Tables

**Figure 1 molecules-25-00889-f001:**
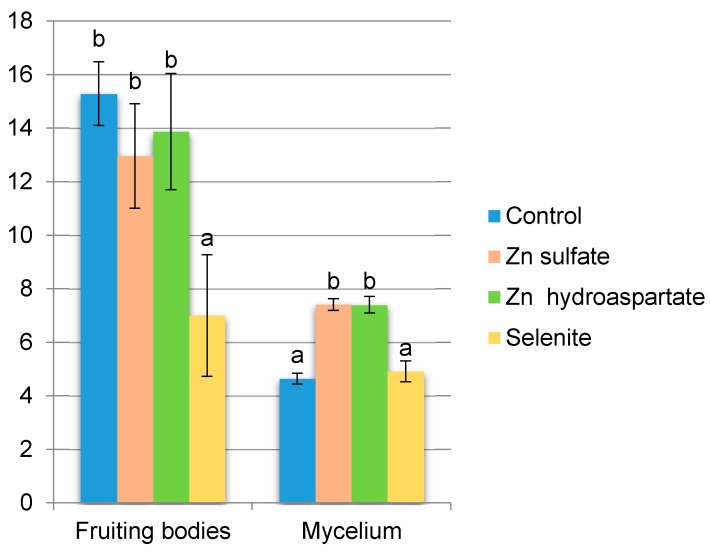
The biomass of *Pleurotus eryngii* fruiting bodies (g of fruiting bodies in d.w. per 100 g^−1^ of d.w. substrate) and mycelium (g of mycelium in d.w. per 1 dm^3^ of liquid Oddoux medium) as dependent on Zn and Se biofortification. Means followed by different letters are significantly different at *p* ≤ 0.05, *n* = 6. Comparisons performed using Tukey’s honestly significant difference test. Error bars represent ± standard deviation (SD).

**Figure 2 molecules-25-00889-f002:**
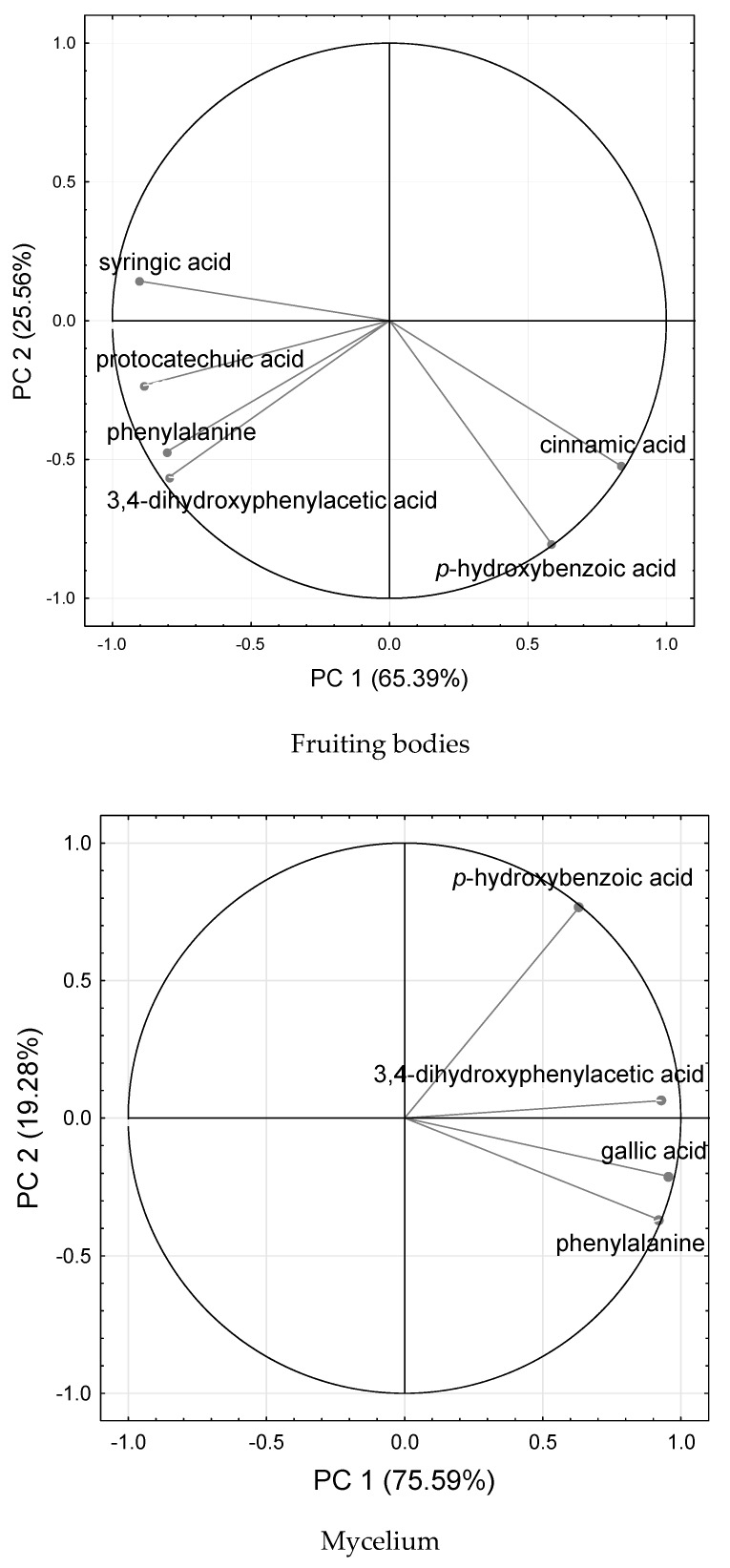
Integrated comparative bi-plot based on principal component analysis (PCA). The position of the organic acids was determined by the first two principal axes (87.9% explained variance for fruit bodies and 94.9% explained variance for mycelium from in vitro cultures).

**Figure 3 molecules-25-00889-f003:**
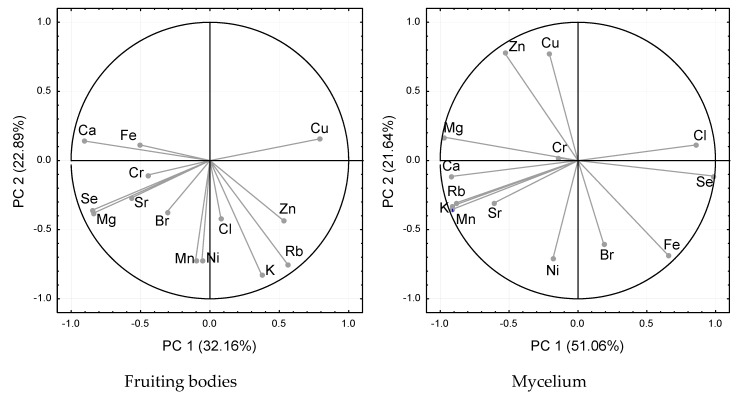
Integrated comparative bi-plot based on principal component analysis (PCA). The position of the elements was determined by the first two principal axes (55.1% explained variance for fruit bodies, 72.7% explained variance for mycelium from in vitro cultures).

**Table 1 molecules-25-00889-t001:** The content of organic acids (mg 100 g^−1^ d.w.), lovastatine (mg 100 g^−1^ d.w.), total phenolics (mg 100 g^−1^ d.w.), and 2,2-Diphenyl-1-picrylhydrazyl (DPPH) scavenging activity (%DPPH˙) in fruiting bodies and mycelium of *P. eryngii* grown in media supplemented with selenite, Zn sulfate, and Zn hydroaspartate.

Treatment	Phenylalanine	Gallic Acid	Protocatechuic Acid	3,4-Dihydroxy Phenylacetic Acid	*p*-Hydroxybenzoic Acid	Syringic Acid	Cinnamic Acid	Lovastatin	Total Phenolics	DPPH˙ Scavenging Activity
	Fruiting Bodies									
Control	176 ± 0.85 ^d1^	n.d.	0.42 ± 0.01 ^b^	99.30 ± 0.38 ^d^	2.44 ± 0.03 ^b^	0.09 ± 0.01 ^b^	0.92 ± 0.01 ^b^	n.d.	212.76 ± 14.04 ^a,b^	33.15 ± 1.03 ^a^
Selenite	141 ± 0.69 ^c^	n.d.	0.40 ± 0.02 ^b^	97.01 ± 0.27 ^c^	2.35 ± 0.03 ^a^	0.09 ± 0.01 ^b^	0.94 ± 0.07 ^a^	n.d.	226.44 ± 25.03 ^b^	39.22 ± 0.47 ^b^
Zn sulfate	137 ± 1.28 ^b^	n.d.	0.36 ± 0.02 ^a^	92.11 ± 0.89 ^b^	3.34 ± 0.03 ^c^	0.06 ± 0.01 ^a^	1.61 ± 0.05 ^d^	n.d.	263.04 ± 10.29 ^c^	41.56 ± 0.75 ^b^
Zn hydroaspartate	121 ± 0.46 ^a^	n.d.	0.36 ± 0.01 ^a^	84.20 ± 0.29 ^a^	2.31 ± 0.02 ^a^	0.07 ± 0.01 ^a^	0.82 ± 0.01 ^a^	n.d.	183.72 ± 14.04 ^a^	33.03 ± 2.21 ^a^
	**Mycelium**									
Control	257 ± 10.4 ^a^	0.58 ± 0.09 ^a^	n.d.	6.75 ± 0.10 ^a^	0.26 ± 0.02 ^b^	n.d.	n.d.	27.02 ± 0.07 ^d^	283.68 ± 12.96 ^a^	54.6 ± 0.25 ^b^
Selenite	622 ± 13.4 ^b^	1.14 ± 0.09 ^b^	n.d.	n.d	0.17 ± 0.01 ^a^	n.d.	n.d.	2.26 ± 0.09 ^c^	257.16 ± 3.11 ^a^	56.5 ± 1.53 ^b^
Zn sulfate	2233 ± 88.9 ^d^	10.42 ± 0.22 ^d^	n.d.	15.78 ± 0.26 ^c^	0.25 ± 0.02 ^b^	n.d.	n.d.	0.11 ± 0.03 ^a^	318.06 ± 3.42 ^a^	39.9 ± 1.37 ^a^
Zn hydroaspartate	1531 ± 45.1 ^c^	8.87 ± 0.26 ^c^	n.d.	8.34 ± 0.07 ^b^	0.28 ± 0.05 ^b^	n.d.	n.d.	0.69 ± 0.04 ^b^	317.52 ± 3.81 ^a^	41.6 ± 0.15 ^a^

^1^ Means in a column followed by different superscript letters, separately for fruiting bodies and mycelium, are significantly different at *p* ≤ 0.05 according to Tukey’s *t* test, *n* = 6. Each value represents the mean ± standard deviation.

**Table 2 molecules-25-00889-t002:** Pearson’s correlation coefficients between Zn and Se (used as supplements for growing media) and organic compounds in fruiting bodies and mycelium of *P. eryngii.*

Organic Compounds	Fruiting Bodies		Mycelium	
	Zn	Se	Zn	Se
Phenylalanine	0.301 *	−0.045	0.766 **	−0.479
Gallic acid	nd	nd	0.890 ***	−0.612 *
Protocatechuic acid	−0.123	0.357	nd	nd
3,4-Dihydroxyphenylacetic acid	0.223	0.397	0.575 *	−0.824 ***
*p*-Hydroxybenzoic acid	0.772 ***	−0.349	0.564 *	−0.842 ***
Syringic acid	−0.273	0.526 *	nd	nd
Cinnamic acid	0.567 *	−0.322	nd	nd
Lovastatin	nd	nd	−0.578 *	−0.176
Total phenolics	−0.738 ***	0.0896	0.195	−0.513 *
2,2-Diphenyl-1-picrylhydrazyl (DPPH) scavenging activity	−0.708 **	0.3555	−0.957 ***	0.711 ***

***, **, * significant at *p* ≤ 0.001, 0.01, and 0.05, respectively. nd—compound not detected.

**Table 3 molecules-25-00889-t003:** The content of bio-elements in cultivation substrate (mg 100 g^−1^ d.w.), fruiting bodies, and mycelium of *P. eryngii* (mg 100 g^−1^ d.w.) grown in media supplemented with selenite, Zn sulfate, and Zn hydro-aspartate.

**Treatment**	**K**	**Mg**	**Ca**	**Zn**	**Fe**	**Cl**	**Rb**	**Cu**	**Se**	**Mn**	**Ni**	**Cr**	**Br**	**Sr**
	**Growth Substrate**													
	449 ± 19	49.8 ± 1.6	153.6 ± 2.3	3.43 ± 0.08	13.6 ± 0.46	1.22 ± 0.33	0.33 ± 0.04	0.27 ± 0.08	0.02 ± 0.02	4.27 ± 0.11	1.24 ± 0.15	1.40 ± 0.25	0.14 ± 0.06	0.84 ± 0.06
	**Fruiting Bodies**													
Control	2328 ± 235 ^c1^	117 ± 7 ^a,b^	17.4 ± 1.6 ^a^	4.94 ± 0.12 ^b^	3.85 ± 0.22 ^a^	1.98 ± 0.36 ^a^	1.64 ± 0.13 ^c^	1.42 ± 0.06 ^c^	0.05 ± 0.05 ^a^	0.90 ± 0.04 ^a^	0.49 ± 0.05 ^b^	0.12 ± 0.02 ^a^	0.10 ± 0.03 ^a^	0.08 ± 0.02 ^a^
Selenite	2070 ± 129 ^b,c^	235 ± 11 ^c^	52.8 ± 4.7 ^d^	4.48 ± 0.35 ^a^	5.11 ± 0.97 ^b^	1.89 ± 0.39 ^a^	1.34 ± 0.06 ^a,b^	0.92 ± 0.03 ^a^	1.36 ± 0.08 ^b^	0.92 ± 0.08 ^a^	0.50 ± 0.11 ^b^	0.16 ± 0.07 ^a^	0.12 ± 0.06 ^a^	0.14 ± 0.09 ^a^
Zn sulfate	1753 ± 240 ^a^	110 ± 2 ^a^	40.1 ± 0.3 ^c^	4.39 ± 0.29 ^a^	4.98 ± 0.37 ^b^	1.85 ± 0.29 ^a^	1.24 ± 0.14 ^a^	1.23 ± 0.23 ^b^	0.03 ± 0.03 ^a^	0.86 ± 0.11 ^a^	0.38 ± 0.05 ^a^	0.13 ± 0.03 ^a^	0.08 ± 0.03 ^a^	0.07 ± 0.02 ^a^
Zn hydroaspartate	2034 ± 122 ^b^	121 ± 3 ^b^	30.5 ± 6.7 ^b^	5.45 ± 0.25 ^c^	5.56 ± 1.06 ^b^	1.86 ± 0.18 ^a^	1.44 ± 0.08 ^b^	1.14 ± 0.09 ^b^	0.03 ± 0.03 ^a^	0.87 ± 0.04 ^a^	0.43 ± 0.06 ^b^	0.11 ± 0.04 ^a^	0.13 ± 0.03 ^a^	0.09 ± 0.03 ^a^
	**Mycelium**													
Control	872 ± 62 ^c^	175 ± 7.1 ^b^	61.8 ± 7.1 ^b^	15.1 ± 0.88 ^b^	31.4 ± 0.36 ^c^	17.46 ± 4.75 ^a,b^	0.29 ± 0.05 ^c^	0.29 ± 0.07 ^a^	2.1 ± 0.03 ^b^	9.98 ± 0.41 ^d^	0.63 ± 0.13 ^b^	0.14 ± 0.06 ^a^	0.26 ± 0.09 ^a^	0.20 ± 0.00 ^a^
Selenite	432 ± 9 ^a^	79 ± 1.5 ^a^	29.1 ± 6.5 ^a^	10.9 ± 0.54 ^a^	33.46 ± 0.88 ^d^	31.64 ± 2.62 ^c^	0.13 ± 0.03 ^a^	0.35 ± 0.07 ^a,b^	18.79 ± 0.44 ^b^	4.46 ± 0.17 ^a^	0.46 ± 0.05 ^a^	0.14 ± 0.02 ^a^	0.24 ± 0.04 ^a^	0.14 ± 0.05 ^a^
Zn sulfate	739 ± 46 ^b^	192 ± 8.2 ^c^	54.7 ± 5.9 ^b^	289.5 ± 1.80 ^d^	22.81 ± 0.74 ^b^	21.95 ± 5.50 ^b^	0.24 ± 0.03 ^b^	0.46 ± 0.06 ^c^	0.23 ± 0.08 ^a^	7.82 ± 0.13 ^b^	0.45 ± 0.08 ^a^	0.17 ± 0.07 ^a^	0.19 ± 0.02 ^a^	0.18 ± 0.03 ^a^
Zn hydroaspartate	729 ± 29 ^b^	193 ± 0.4 ^c^	61.8 ± 6.7 ^b^	178.3 ± 2.43 ^d^	20.74 ± 1.28 ^a^	16.09 ± 1.46 ^a^	0.25 ± 0.00 ^b,c^	0.41 ± 0.03 ^b,c^	0.36 ± 0.05 ^a^	8.48 ± 0.21 ^c^	0.41 ± 0.07 ^a^	0.12 ± 0.06 ^a^	0.19 ± 0.02 ^a^	0.18 ± 0.03 ^a^

^1^ Means in a column followed by different superscript letters, separately for fruiting bodies and mycelium, are significantly different at *p* ≤ 0.05 according to Tukey’s *t* test, n = 6. Each value represents the mean ± standard deviation.

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
