# Peer review of "Selenium and Zinc Biofortification of Pleurotus eryngii Mycelium and Fruiting Bodies as a Tool for Controlling Their Biological Activity"

_molecules, 2020, doi:10.3390/molecules25040889_

Round 1

Reviewer 1 Report

Results and discussion section is poor. Please provide the scientific reason for the behavior of the parameters

In the materials part, the authors should report the name of the chemical company, countries and, write that the chemicals were used with or without purification

Discuss in brief about optimization of process variables in context to methodology aspects.

The resolution of all Figures should be improved.

The typographical, punctuation and grammatical mistakes should be corrected throughout the text

The authors need to cite some more recent papers published in this area

Author Response

We would like to thank the Reviewers for their thoughtful comments and efforts towards improving our manuscript. We address all comments below, and we marked changes in the text of manuscript using the "Track Changes" function of Microsoft Word.

With best regards, 

Piotr Zięba MSc and Co-authors

Our responses to the reviewer’s comments are as follows:

Thank you for evaluating our manuscript. We hope that we were able to address each of your concerns.

Authors' Responses to Reviewer 1 comments 

Reviewer 1

Results and discussion section is poor. Please provide the scientific reason for the behavior of the parameters.

Authors’ Response:

Thank you for this suggestion. We clarified and supplemented both mentioned chapters and discussion as well,  with the scientific validation of the  results. We carefully reviewed the manuscript and we hope that we suitably addressed your comment.

Reviewer 1

In the materials part, the authors should report the name of the chemical company, countries and, write that the chemicals were used with or without purification

Authors’ Response:

Thank you for this suggestion. We clarified and supplemented both mentioned chapters and discussion as well,  with the scientific validation of the  results. We carefully reviewed the manuscript and we hope that we suitably addressed your comment.

Reviewer 1

Discuss in brief about optimization of process variables in context to methodology aspects.

Authors’ Response:

We supplemented discussion  in brief conclusions focused on linking experimental variables to predicted results of the processes described.

Reviewer 1

The resolution of all Figures should be improved.

Authors’ Response:

We uploaded high-resolution images separately, as ZIP file.

Reviewer 1

The typographical, punctuation and grammatical mistakes should be corrected throughout the text

Authors’ Response:

We corrected manuscript regarding typographical, punctuation and grammatical errors.

Reviewer 1

The authors need to cite some more recent papers published in this area.

Authors’ Response:

We included more recent and the most relevant references in Introduction and Discussion chapters.

Reviewer 2 Report

The paper explores the question of whether increased zinc and selenium doses in the nutrient medium lead to biofortification of the corresponding trace elements in the mushroom P. eryngii. Different doses of zinc and selenium are administered in the form of different salts, but it is not entirely clear which salts have been added. The correct chemical names according to IUPAC should be given for all suplemented salts e.g. zinc sulfate (is it ZnSO4 x 7 H2O or ZnSO4 x H2O) and what is zinc hydroaspartate. There are some more inaccuracies in the section materials and methods: All concentration should be given in mol for a better comparison with reference values. It should be mL and not cm3, e.g. Calcium chloride hexahydrate is CaCl2 x 6 H2O and so on for other salts. The total phenol content is determined by Folin Ciocalteu method and the results are expressed as gallic acids equivalents in mg/g. For the bioelements analysis using TXRF method validation parameter should be given e.g. LOD, LOQ and recovery rates.

Author Response

We would like to thank the Reviewers for their thoughtful comments and efforts towards improving our manuscript. We address all comments below, and we marked changes in the text of manuscript using the "Track Changes" function of Microsoft Word.

With best regards, 

Piotr Zięba MSc and Co-authors

Our responses to the reviewer’s comments are as follows:

Thank you for evaluating our manuscript. We hope that we were able to address each of your concerns.

Authors' Responses to Reviewer 2 comments 

Reviewer 2

The paper explores the question of whether increased zinc and selenium doses in the nutrient medium lead to biofortification of the corresponding trace elements in the mushroom P. eryngii. Different doses of zinc and selenium are administered in the form of different salts, but it is not entirely clear which salts have been added.

The correct chemical names according to IUPAC should be given for all suplemented salts e.g. zinc sulfate (is it ZnSO4 x 7 H2O or ZnSO4 x H2O) and what is zinc hydroaspartate.

Authors’ Response:

We corrected chemical nomenclature of salts used for experiments and used as reagents in analyses and we added the concentrations of salts used for fortification of in vitro cultures and cultivation at a concentration of 100 mg L−1 : zinc sulfate (0.000304 mol L−1); zinc hydroaspartate (0.00027 mol/L); sodium selenite – 0.000289 mol/L

Reviewer 2

There are some more inaccuracies in the section materials and methods: All concentration should be given in mol for a better comparison with reference values. It should be mL and not cm3, e.g. Calcium chloride hexahydrate is CaCl2 x 6 H2O and so on for other salts.

Authors’ Response:

We corrected cm3 to mL, and chemical names as we stated above. The reagents are contained in Oddoux medium (Oddoux ,1957).

The total phenol content is determined by Folin Ciocalteu method and the results are expressed as gallic acids equivalents in mg/g. For the bioelements analysis using TXRF method validation parameter should be given e.g. LOD, LOQ and recovery rates.

Authors’ Response:

We have shown total phenol content determined by Folin Ciocalteu method in mg per 100 g−1 d.w. similar to the other organic compounds presented in table 1.

We added: The highest detection limits (LOD) are observed for potassium and calcium above 1 mg/kg for light elements, the remaining LOD values oscillate below 1 mg/g (0; 0.09; 012; 0.22; 0.29; 0.38; 0.38; 0.45; 046, 0.99 for Rb, Se, Sr, Ni, Cu, Zn, Fe, Mn, Cr, Mg respectively). 

Reviewer 3 Report

The manuscript entittled "Selenium and zinc biofortification of Pleurotus eryngii mycelium and fruiting bodies as tool to controlling theirs biological activity" is very well presented and described. The research seems to be a nice starting point for new research in the field of mushrooms biofortification.

Some suggestions are made in the attached document to improve the presentation of the data and some doubts/recommendations are given in comments.

My main recommendation is to explain, and include in the introduction section, why the authors use mycelium for analysis. They talk about the commercial part (the fruiting bodies), but they don't justify anywhere why they use mycelium. They should explain the benefit to improve minerals or organic compounds in the mycelium. In addition, as shown in the attached document, I suggest to improve discussion section, especially when talking about mineral absorption; some important information is missing under my point of view.

Author Response

We would like to thank the Reviewers for their thoughtful comments and efforts towards improving our manuscript. We address all comments below, and we marked changes in the text of manuscript using the "Track Changes" function of Microsoft Word.

With best regards, 

Piotr Zięba MSc and Co-authors

Our responses to the reviewer’s comments are as follows:

Thank you for evaluating our manuscript. We hope that we were able to address each of your concerns.

Authors' Responses to Reviewer 3 comments 

Reviewer  3

The manuscript entittled "Selenium and zinc biofortification of Pleurotus eryngii mycelium and fruiting bodies as tool to controlling theirs biological activity" is very well presented and described. The research seems to be a nice starting point for new research in the field of mushrooms biofortification.

Authors’ Response:

Thank you very much for positive general opinion on the manuscript.

Reviewer 3

Some suggestions are made in the attached document to improve the presentation of the data and some doubts/recommendations are given in comments.

Authors’ Response:

Thank you for all detailed comments, we carefully corrected the manuscript according to your suggestions.

Reviewer 3

My main recommendation is to explain, and include in the introduction section, why the authors use mycelium for analysis. They talk about the commercial part (the fruiting bodies), but they don't justify anywhere why they use mycelium. They should explain the benefit to improve minerals or organic compounds in the mycelium.

Authors’ Response:

Thank you for this suggestion. We include a separate subchapter to the Introduction to explain the grounds for using mycelium as source bio-chemicals.

Reviewer 3

In addition, as shown in the attached document, I suggest to improve discussion section, especially when talking about mineral absorption; some important information is missing under my point of view.

Authors’ Response:

We improved the discussion to highlight the advantages and a disadvantages of mushroom supplementation with Zn and Se salts.